# Unmasking the Veil: An Investigation into Concept Ablation for Privacy and Copyright Protection in Images

**Shivank Garg**[*]                                                      *shivank_g@mfs.iitr.ac.in*
*Department of Data Science and Artificial Intelligence*
*IIT Roorkee*

**Manyana Tiwari**[*]                                                     *m_tiwari@ma.iitr.ac.in*
*Department of Mathematics*
*IIT Roorkee*

**Reviewed on OpenReview:** *https://openreview.net/forum?id=TYYApLzjaQ*

## Abstract

In this paper, we extend the study of concept ablation within pre-trained models as introduced in 'Ablating Concepts in Text-to-Image Diffusion Models' by (Kumari et al., 2022). Our work focuses on reproducing the results achieved by the different variants of concept ablation proposed and validated through predefined metrics. We also introduce a novel variant of concept ablation, namely 'trademark ablation'. This variant combines the principles of memorization and instance ablation to tackle the nuanced influence of proprietary or branded elements in model outputs. Further, our research contributions include an observational analysis of the model's limitations. Moreover, we investigate the model's behavior in response to ablation leakage-inducing prompts, which aim to indirectly ablate concepts, revealing insights into the model's resilience and adaptability. We also observe the model's performance degradation on images generated by concepts far from its target ablation concept, documented in the appendix.

## 1 Introduction

With the advent of sophisticated large-scale text-to-image diffusion models, there have been significant improvements in the quality and compositional ability of the generated images. This is attributed to the utilization of extensive training data, which often contains copyrighted material and licensed images. Therefore, these models have the ability to replicate the styles of various artists or memorize exact training samples. Hence, there is an imminent need to develop techniques to unlearn(ablate) the effect of a specific "target concept" from the output generated by the model.

**Concept ablation** is defined as the task of preventing the generation of the desired image corresponding to a given target concept that needs to be ablated. This is done by modifying the generated images to match the image distribution corresponding to a more generalized anchor text prompt instead of the specific target prompt. A simple approach maximizes the diffusion training loss for the target prompt-image pairs. However, this produces poor relevance to the "anchor concept" and results on the generated images.

The recent paper 'Ablating Concepts in Text-to-Image Diffusion Models' Kumari et al. (2023) provided an efficient method for ablating concepts from a pre-trained model without retraining the model from scratch. Their work aims to improve the degree of retention of the anchor and surrounding concepts while ablating a target concept.

Through this paper, we have attempted to reproduce the original findings and produce new insights and metrics to analyze the accuracy of the proposed method. We have introduced a new variant of ablation,

---

[*]Equal Contribution

namely trademark ablation. We discuss the results and motivation behind this variant in later sections. We also study the robustness of the method in circumventing leakage-inducing prompts. Our observations also showcase that the model performs sub-optimally on ablating concepts far from the anchor concepts.

**Paper Outline** Through Section 2 we build a background of the working of the proposed ablation models. We provide details of training and baseline metrics in Section 3. Further, in Section 4, we expand on the analysis of our variation and limitations of the proposed algorithm.

## 2 Model breakdown and Intuition

The major contribution of the original paper is significantly reducing the latency to ablate a specific target prompt to under 5 minutes. For this, the authors have utilized a loss function based on minimizing the KL Divergence between the distribution of the target concept image (to be ablated) and the distribution of the anchor concept images (Kumari et al., 2022)

### 2.1 Methods

The authors have introduced two different methods for ablation, namely noise-based and model-based ablation. After evaluation on both methods, it was observed that the noise-based method achieves comparable performance to the model-based ablation in optimal cases, and sub-par in some cases. It also has a slower convergence rate overall, thus being an inferior method. Hence, all results in this study are computed solely on the model-based ablation method.
Breaking down the notations used in the following equations:

- $D_{KL}$ is the KL Divergence, which we aim to optimize.

- $p(X_{(0...T)}|c)$ is the distribution of the target concept conditioned on an anchor concept $c$

- $p_{\hat{\Phi}}(X_{(0...T)}|c^*)$ is the model's distribution of the target concept conditioned on a set of text prompts $c*$

- $\alpha_t$ is the strength of gaussian noise $\epsilon$.

- $x_t = \sqrt{\alpha_t x} + \sqrt{1 - \alpha_t \epsilon}$ is the intermediate latent image with iterative noise addition.

- $\hat{\Phi}(\mathbf{x}_t, \mathbf{c}*, t)$ is the denoising network that generates $x_{t-1}$; whose parameters we aim to estimate.

#### 2.1.1 Naive Approach

It is essential to mention the standard diffusion training objective that was introduced as a naive approach to performing concept ablation. This objective maximized the diffusion loss on the text-image pairs corresponding to the target concept, while regularizing the weights. (Kong & Chaudhuri, 2023)

$$\arg \min_{\hat{\Phi}} D_{KL} \left( p(X_{(0...T)}|c) \parallel p_{\hat{\Phi}}(X_{(0...T)}|c^*) \right) \tag{1}$$

#### 2.1.2 Noise-based Ablation

The anchor distribution can be redefined as the pairing of the target prompt and the anchor prompt. An image, denoted by x, is created using the anchor prompt. Introducing randomly sampled noise to x, represented by $\epsilon$, produces a noisy image $x_t$ at time-step $t$. The model is then fine-tuned using these redefined target-anchor pairs, employing the standard diffusion training loss.

$$\mathcal{L}_{\text{noise}}(\mathbf{x}, \mathbf{c}, \mathbf{c}^*) = \mathbb{E}_{\epsilon, \mathbf{x}, \mathbf{c}^*, t}[w_t || \epsilon - \hat{\Phi}(\mathbf{x}_t, \mathbf{c}^*, t)||] \tag{2}$$

### 2.1.3 Model-based Ablation

The method proposed is similar to the standard diffusion model training objective. The authors show that minimizing the KL divergence objective between the joint distribution of noisy latent variables conditioned on anchor and target concept, i.e. loss in the main paper, can be reduced to the difference between the predicted noise vectors. This reduced form is then used for training. (Kumari et al., 2023)

$$D_{KL}\left(p(X_{(0\ldots T)}|c) \parallel p_{\hat{\Phi}}(X_{(0\ldots T)}|c^*)\right) \tag{3}$$

Reduces to :

$$\arg\min_{\hat{\Phi}} \mathbb{E}_{\epsilon,\mathbf{x}_t,\hat{\mathbf{c}},\mathbf{c},t}[w_t \parallel \Phi(\mathbf{x}_t,\mathbf{c},t) - \hat{\Phi}(\mathbf{x}_t,\mathbf{c}*,t)\parallel], \tag{4}$$

A similar modified loss function is also used, which minimizes the reverse KL Divergence instead. Since the reduced form for both the losses is the same (i.e., the difference in noise vectors), they produce sufficiently similar results.

However, this requires extensive memory and computation. To bypass this, we sample $x_t$ using the forward diffusion process and assume that the model remains similar for the anchor concept during fine-tuning. Therefore, we use the network $\hat{\Phi}$ with *stopgrad* to get the anchor concept prediction. Thus, our final training objective is:

$$\mathcal{L}_{\text{model}}(\mathbf{x},\mathbf{c},\hat{\mathbf{c}}) = \mathbb{E}_{\epsilon,\mathbf{x},\hat{\mathbf{c}},\mathbf{c},t}[w_t \parallel \hat{\Phi}(\mathbf{x}_t,\mathbf{c},t) \cdot \text{sg}() - \hat{\Phi}(\mathbf{x}_t,\hat{\mathbf{c}},t)\parallel], \tag{5}$$

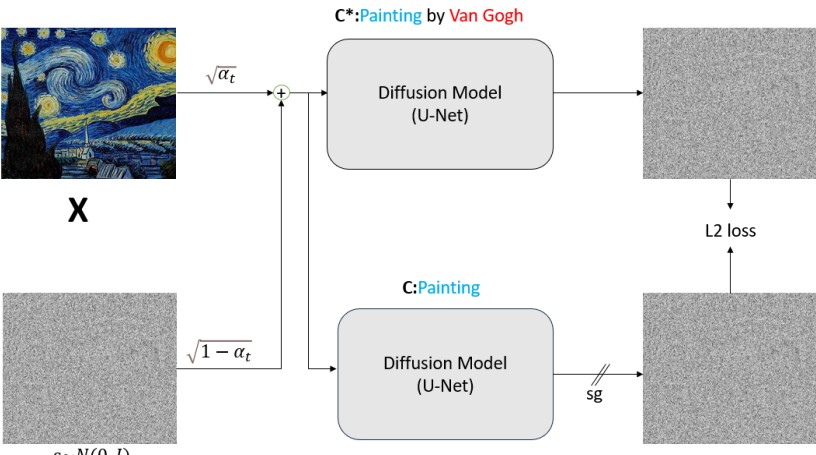

Figure 1: The method for fine-tuning the model involves adjusting the model weights to alter the distribution of generated images related to the target concept. The distribution for the anchor images is created from the diffusion model itself, with conditioning on the anchor concept. For instance, when removing Van Gogh's style, the goal is to align its distribution with the distribution of generic paintings in the diffusion model.

Through the following graphs, we provide a comparative study between the performances of model-based and noise-based ablation methods.

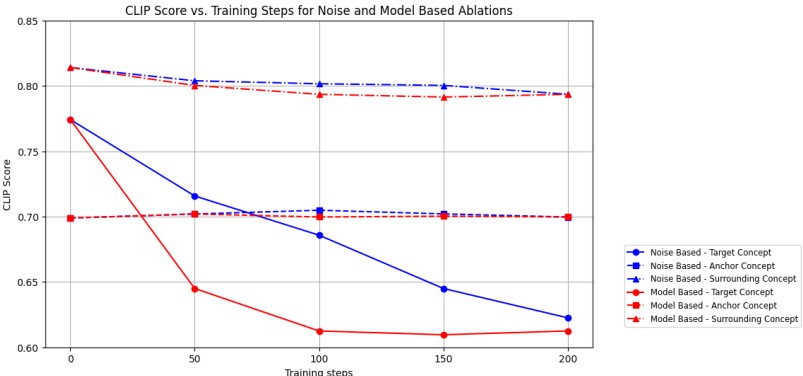

Figure 2: A comparison of the proposed methods, specific to Object Ablation. The resulting CLIP scores are based on ablating the concept "grumpy cat".

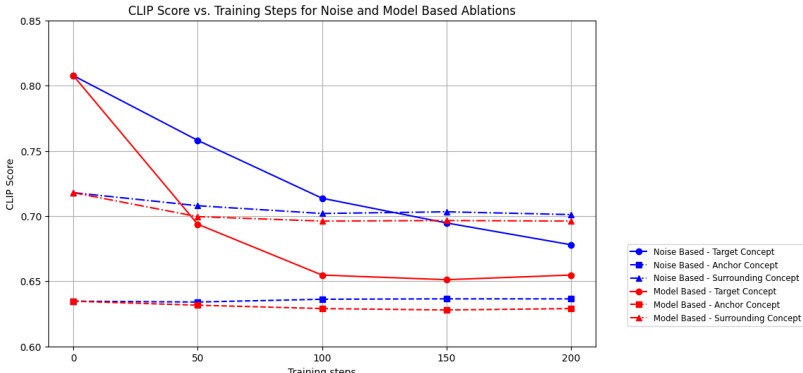

Figure 3: A comparison of the proposed methods, specific to Style Ablation. The resulting CLIP Scores are based on ablating the concept "van gogh".

From the preceding observations, we are able to prove empirically that Noise-based ablation is worse off than Model-based ablation method, quantified by trends in CLIP scores for both style and instance ablation cases. We can view a sharp decline in CLIP scores of the target concept for Model-based ablation, thus validating the claims made by the authors. Hence, we have used the Model-based variant itself substantiate our study further.

## 3 Reproducibility Results

In this section, we report the results obtained on reproducing the methods of the original paper. We have documented the ablation results for the three different variants proposed and added visuals and analysis for all of them, some of which are not included in the original paper.

The variants are:

- **Style:** The motivation behind this ablation variant is to remove an artist's specific influence from an image. This focuses on ablating certain specific textures and patterns.

- **Instance:** This variant aims to remove specific object instances or events within images.

- **Memorized Images:** This variant memorizes and ablates exact recurring training data concepts.

### 3.1 Training Details

All the standard training parameters are documented here.

- **Prompt Generation** We used ChatGPT 4 (ChatGPT) to generate 200 random prompts that contain the anchor prompt and 10 separate evaluation prompts. The prompts we used followed the format:

  - Training prompts: "Give 200 prompts for a text-to-image model containing <anchor-concept> and ensure that the prompts mention <anchor-concept> explicitly. Also, very few of these should contain <target-concept>".

  We ensured the prompts were descriptive and generic enough to avoid extensive emphasis on the target concept, which might mislead results by overfitting the model.

  - Evaluation prompts: "Give 10 prompts for a text-to-image model containing <concept-type>", where the <concept-type> took up values <target-concept>, <anchor-concept>, <surrounding-concept>.

- **Baseline Image Generation** We use the pre-trained checkpoint of the Stable Diffusion model (ckpt v1-4) (Rombach et al., 2022) to generate 200 images for baseline comparison on the same set of prompts.

- **Fine-tune Parameters** We have reproduced results by fine-tuning the following architectural components of the diffusion model.

  - Cross-Attention: The key and value projection matrices in the diffusion model's U-Net. (Kumari et al., 2022)
  - Embedding: The text embedding in the text transformer. (Gal et al., 2022)
  - Full Weights: All parameters of the U-Net. (Ruiz et al., 2022)

### 3.2 Evaluation Metrics

We have utilized the CLIP Score as a quantitative metric to assess the model performance. The CLIP Score provides a value to establish relevance between the prompt and the image generated from it. This is computed by minimizing the distance between matching image-text pairs and maximizing it for non-matching ones using a contrastive loss function. (Hessel et al., 2022). We have used the CLIP-vit-base-patch16 version of the CLIP Model. (Source code)

In the following section, we provide the results of our training corresponding to each concept ablation variant.

### 3.3 Evaluation Results

We computed CLIP scores for 10 different surrounding concepts. The surrounding concepts were chosen manually based on the image and prompts. For each of the Style and Instance variants, the final CLIP score is an average of 25 image scores, and for the Memorization variant it is an average of 10 image scores.

**Note:** Our purpose for reproducing was to showcase the results with an emphasis on trends rather than the numerical accuracy of the scores. The graphs for Memorization ablation are based on standard CLIP score calculation, done without normalization (which was used by the authors). Hence, the CLIP scores are slightly deflated. (Zhengwentai, 2023) However, in order to reproduce results for style and instance ablation, we have used the normalized version of the CLIP Score, as used by the authors.

#### 3.3.1 Style Variant

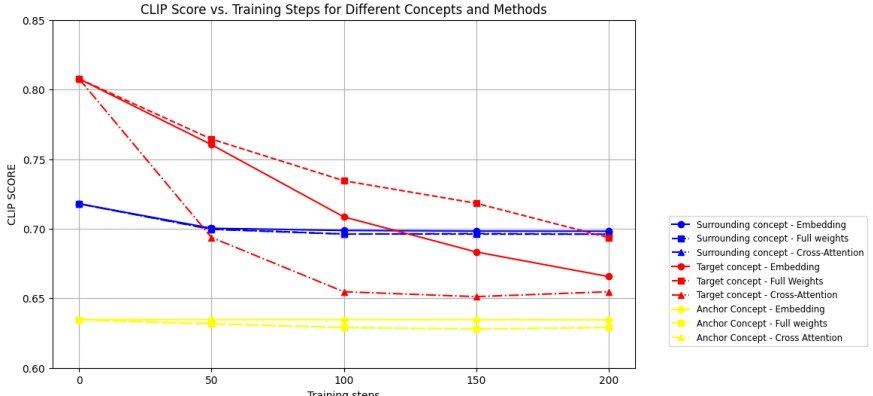

Figure 4: CLIP Scores for Style Ablation. We ablated images with the target concept "Van Gogh" and the anchor concept "Painting". We evaluated it on surrounding concepts of other artists to show the robustness of the model.

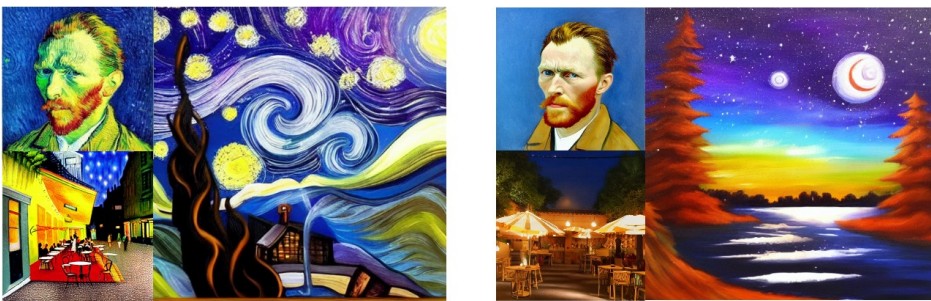

Figure 5: Example of ablation of Van Gogh Style. *Left:* Images produced by the baseline diffusion model. *Right:* Images produced by ablated model

### 3.3.2 Instance Variant

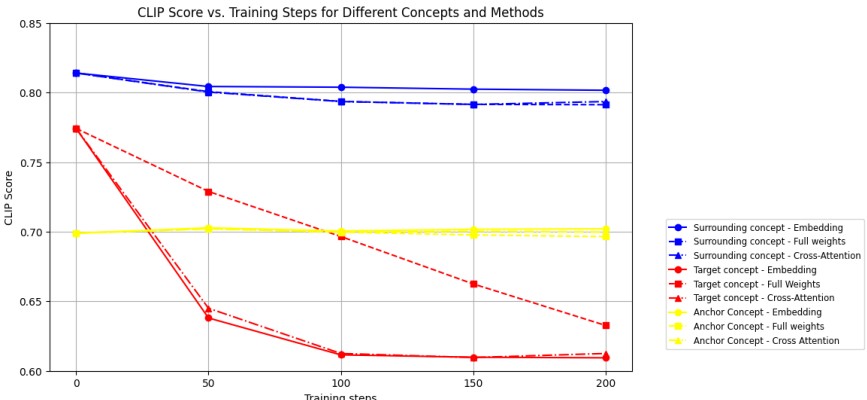

Figure 6: CLIP Scores for Instance Ablation. We ablated images with the target concept "Cat+Grumpy Cat", and the anchor concept being "cat".

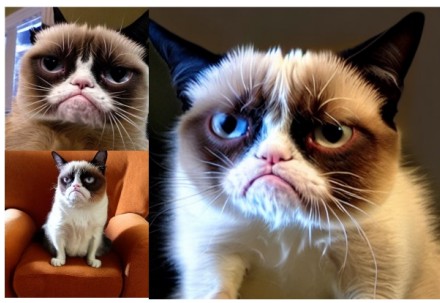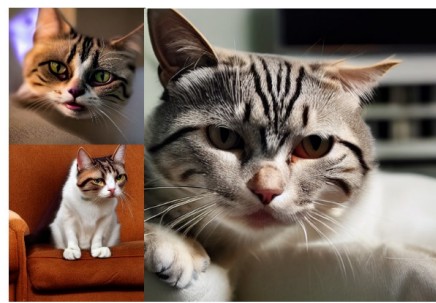

Figure 7: Example of ablation of Grumpy Cat Instance. *Left:* Images produced by baseline diffusion model. *Right:* Images produced by ablated model

### 3.3.3 Memorization Variant

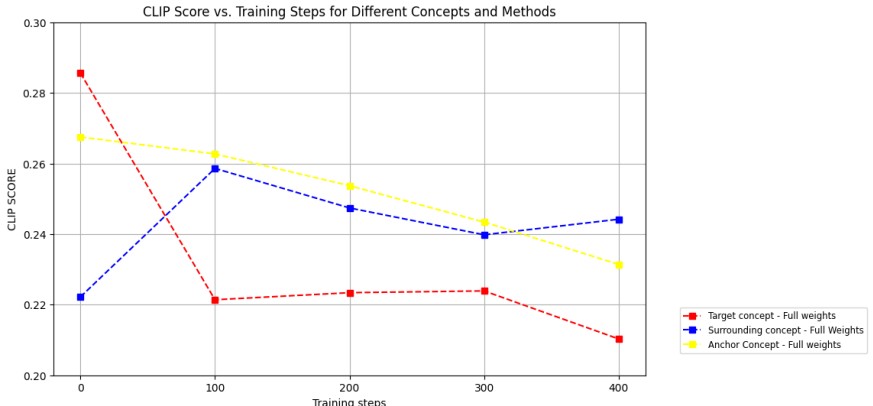

Figure 8: CLIP Scores for Memorization Ablation.

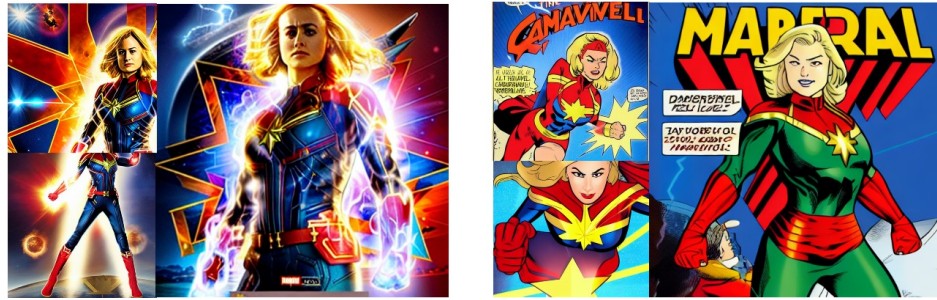

Figure 9: Example of memorized ablation. *Left:* Images produced by baseline diffusion model. *Right:* Images produced by ablated model

## 4 Additional Analysis and Limitations

### 4.1 Trademark Ablation

We propose a new variant of ablation based on the limitation we observed after performing experiments on the three categories.

### 4.1.1 Motivation

While experimenting with instance ablation, we observed a limitation, that the method could not completely ablate certain symbols of well-known brands/trademarks on objects. This undermines the vision for ablation since it is still visually possible to determine the proprietary information about the object. To tackle this, we proposed combining the hyperparameters of Memorization ablation with the basic configuration of Instance ablation.

### 4.1.2 Proposed Method

We have modeled our method on the instance ablation variant, introducing the following changes to its configuration:

1. **Freezing parameters**: We changed the set of parameters to be fine-tuned, fixing them to "all parameters" of the U-Net.(Ruiz et al., 2022)

2. **Augmentations**: We have restricted augmenting the images, since we require the model to memorize the specific orientation and features of the target trademark concept.

3. **Evaluation**: We will evaluate this variant on a generic anchor concept, "logo". This restrains any overfitting and generalizes the anchoring instead of emphasizing alternate trademarks.

### 4.1.3 Experiment Results

The results of the trademark variant support our claims of better ablation of the specific object. We achieved a lower CLIP score for the target concept, showcasing better target ablation.

However, we are not able to observe a drastic decrease in the CLIP score despite the complete visual absence of the logo. This is because CLIP compares the overall image with the entire prompt. (Radford et al., 2021) Therefore, the image retains a large semantic similarity with the prompt, even without the presence of the trademark logo.

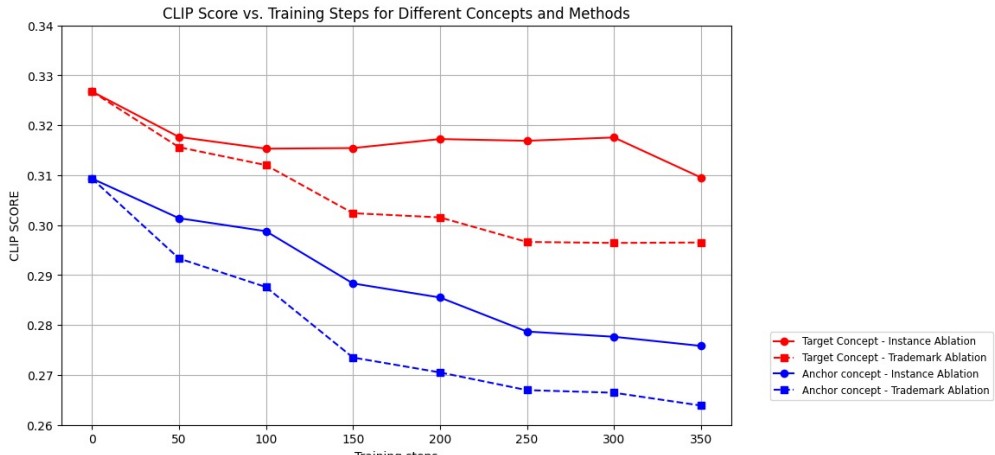

Figure 10: CLIP Scores after finetuning on our trademark variant configuration. We can observe a lowered CLIP score for the target concept, indicating better ablation. However, we can also observe a lowered CLIP score for the anchor concept. We hypothesize this is due to the model learning to ablate the entire logo/trademark class in the process of ablating a proprietary trademark symbol.

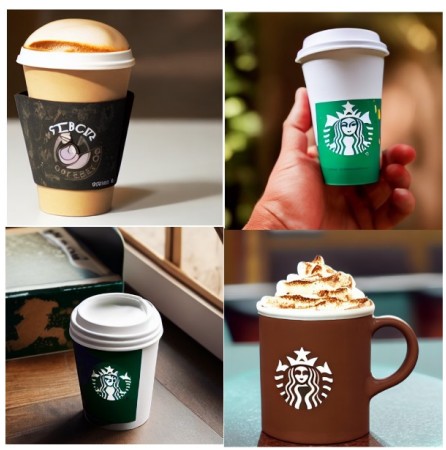 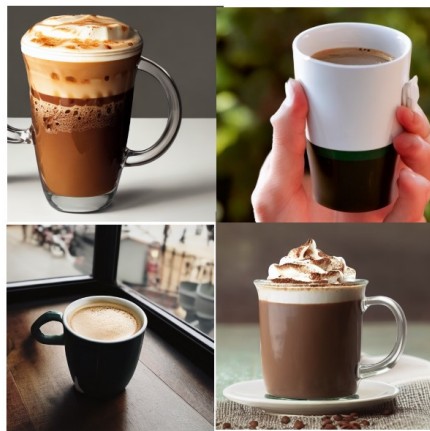

Figure 11: Examples of complete ablation of the Starbucks logo. We kept the number of training steps and the global seed the same for both variants to ensure the fairness of the experiment. *Left:* Generated by the instance ablated model. *Right:* Generated by our trademark ablation model.

## 4.2 Ablation leakage

An important metric to judge the success of ablation is if it is able to circumvent indirect or leakage-inducing target prompts, i.e., when the prompt does not explicitly include the target concept but describes it in similar terms, which induces the model into generating it. We showcase the inability of the proposed methods to prevent ablation in such cases. We evaluated the performance of the variants by sampling generated images and contrasting them with that of the pre-trained model.

### 4.2.1 Instance and Style variant

We observed that the instance and style variants are relatively leakage-resistant. For example, while observing the performance after ablating the target concept of "R2D2", we were not able to generate images that contained the r2d2 concept despite using prompts such as "Visualize a smart and brave astromech droid with a white and blue design, a flat base with two legs, and a head that resembles a radar dish."

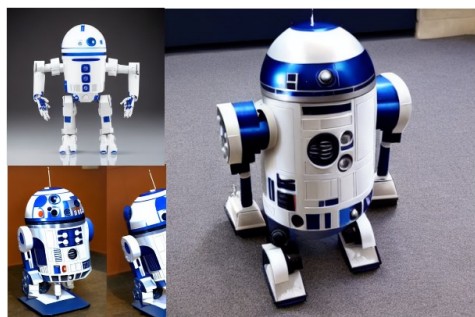 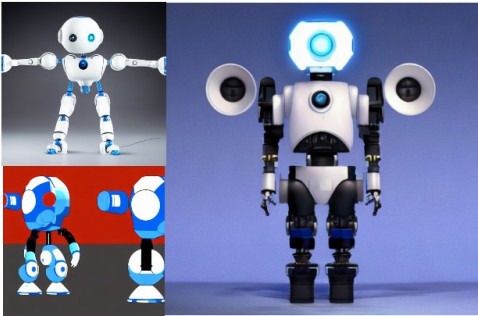

Figure 12: Leakage prompt evaluation on r2d2. *Left:* Generated by baseline stable diffusion model. *Right:* Generated by instance ablated variant.

Similarly, while observing the performance after ablating van gogh style, we used prompts such as "Cafe terrace at night" and "Vase with 15 sunflowers". However, the model ceased to generate van gogh style.

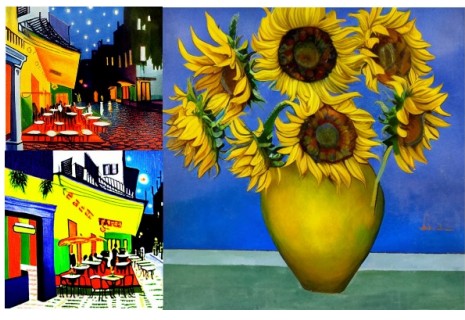 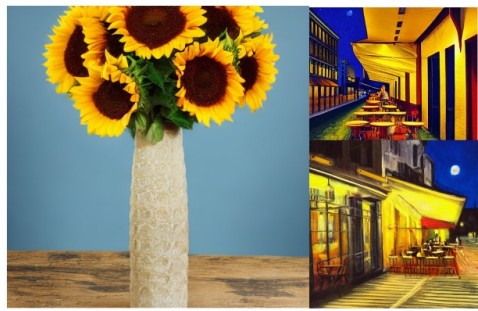

Figure 13: Leakage prompt evaluation on van gogh paintings. *Left:* Generated by baseline stable diffusion model. *Right:* Generated by style ablated variant.

### 4.2.2 Memorization variant

In comparison, we could induce the memorization ablated model to produce a target concept of "bitcoin", even with vague prompts such as "Cryptocurrency investments can be highly volatile, with prices fluctuating rapidly". (Zhang et al., 2023b)

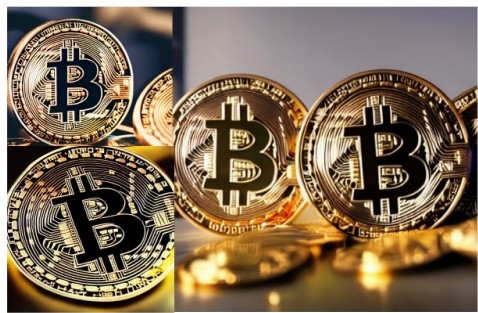 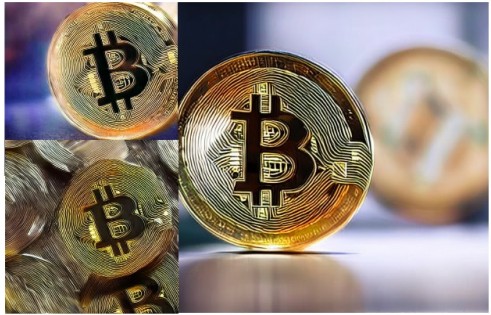

Figure 14: *Left*: Images generated by the baseline *Right:* Images generated by ablated model.

## 5 Discussion

In this section, we further hypothesize reasons for the limitations of the proposed methods.

- The degradation of surrounding concepts whilst ablating a target concept can be due to similar probability distributions. This similarity can be verified by examining the method of encoding tokens. While training, the shift of the target distribution to the anchor distribution might also result in a similar shift for surrounding concepts. (Gandikota et al., 2023)

We also provide some strategies that could be useful in mitigating the listed limitations, specifically the degraded performance of far-away concepts. This includes additional optimizations. We propose utilizing the following paths:

- The decision boundaries of classification tasks can be used to generate rehearsal samples that are more susceptible to forgetting. Retraining the model on those examples will ensure better performance on far-away concepts. (Cywiński et al., 2024)

- Guided diffusion processes, such as the ControlNets can be used to supervise the output produced by the diffusion model, and guide the output using conditional losses. (Zhang et al., 2023a).

**Note:** The CLIP Score is a skewed metric for verifying the absolute performance of a concept ablation model. This is due to the reasoning that the CLIP compares the entire textual prompt with the image. Hence, the reasoning for a dip in the CLIP score cannot be attributed solely to the success of ablation. This is why we cannot determine a certain threshold below which we can deem a model successful. (Stein et al., 2023)

## 6 Conclusion

Through the course of this work, we have reproduced the claims made by the authors, along with providing additional visual metrics to analyze the results. We experimented with a new variant of concept ablation, the trademark variant. We showcased its performance superseding that of instance ablation on the specific objective of completely ablating proprietary identification symbols on objects. We also showcased various limitations of the proposed methods.

**Broader Impact Statement**

The work of this paper encourages thought on exploring the ablation of proprietory objects and products. This provides institutions with the choice of retaining the privacy of their products.

**Acknowledgements**

We would like to thank Aniket Agarwal for providing invaluable guidance throughout, as well as the members of VLG IITR for providing valuable feedback and participating in the user study. We would also like to thank the authors of the original work for making their code publicly available, allowing us to conduct our experiments effectively.

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

# A  Appendix

Visual observations of our experimentations.

## A.1  Image generation for a far-off concept

We provide some observations on the degrading quality of images generated by a model that ablates a far-off concept.

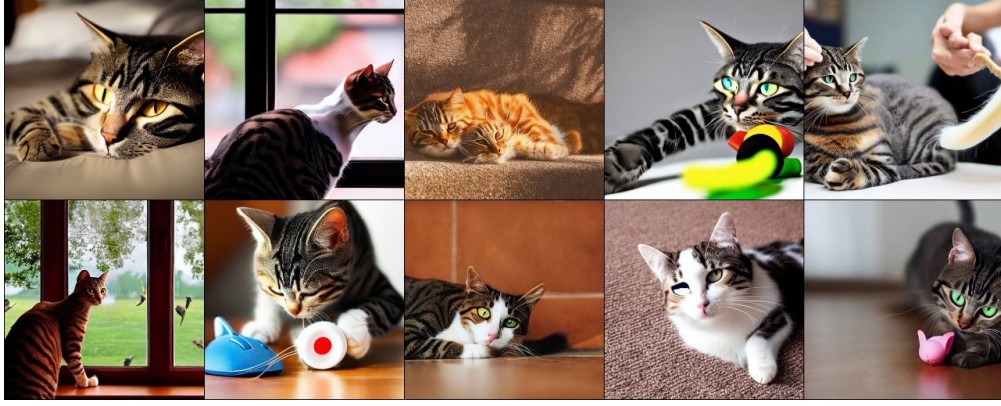

Figure 15: The model here was trained to ablate the target instance "Snoopy Dog". We observe the poor compositional ability of the model to generate images of cats.

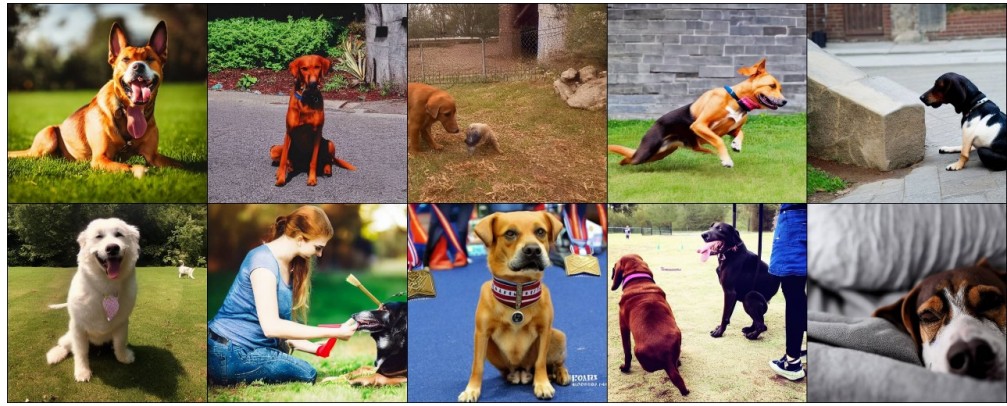

Figure 16: The model here was trained to ablate "Grumpy Cat". While generating images of an unrelated concept of "dogs", we observe poor compositional ability.

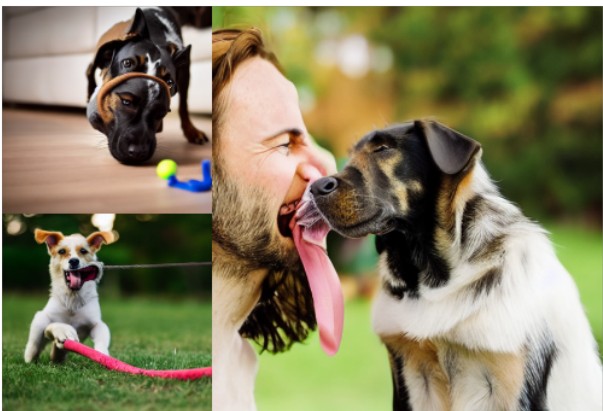

Figure 17: Some more images of the above example where we observe poor compositional ability.

## A.2   Trademark Ablation Examples

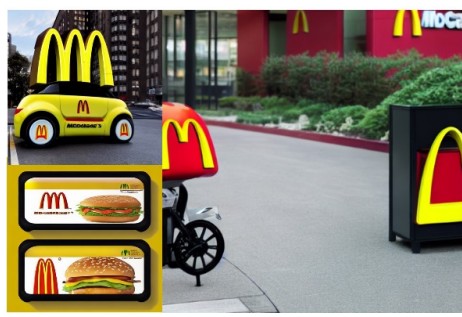 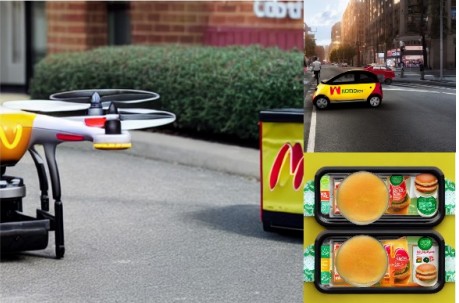

Figure 18: Our variant of the trademark ablated model ablated the target concept of the McDonald's Logo. We get promising results, as shown.

## A.3   Misc Generated Examples

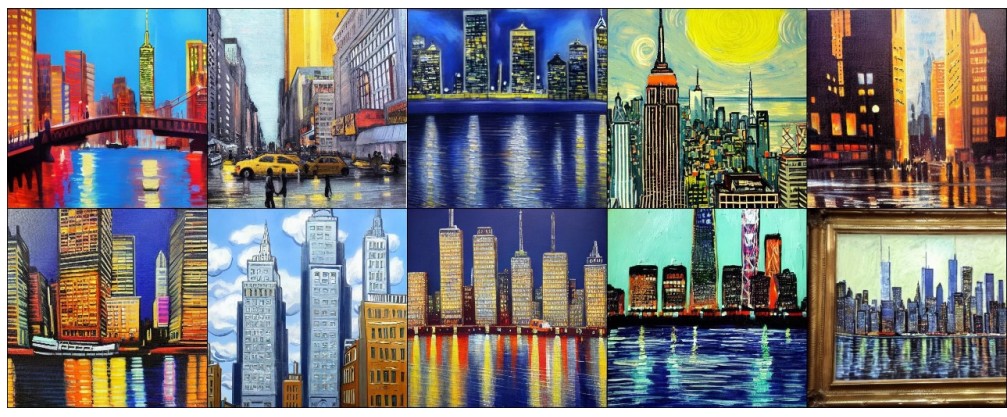

Figure 19: The model was trained to ablate Van Gogh style, on sampling various images for the prompt "New York in Van Gogh style", we get the above results

## A.4 User Study

In order to support the legibility of our claims, we performed a user study. The participants were asked to choose between pairs of images, one belonging to our Trademark ablation model and the other to the proposed Instance ablation model. The orientation of our images was shuffled constantly between left and right, ensuring no prior bias. We performed this study on 33 participants.

### A.4.1 Results

Taking the average of feedback from all participants:

- For the ablated concept, the metric of choice is the success of ablation, given the target prompt. We evaluated these on a set of 45 pairs of images:

    - Trademark ablation better: $38/45 \approx 84.5\%$
    - Similar performance: $6/45 \approx 13.3\%$
    - Instance ablation better: $1/45 \approx 2.2\%$

- For the anchor concept, the metric of choice is the compositional ability, given the anchor concept. We further added 20 pairs for testing:

    - Trademark ablation better: $46/65 \approx 70.8\%$
    - Similar performance: $12/65 \approx 18.5\%$
    - Instance ablation better: $7/65 \approx 10.7\%$

