# OpenReview forum: "Unmasking the Veil: An Investigation into Concept Ablation for Privacy and Copyright Protection in Images"
_TMLR — Accepted by TMLR_

### Review · Reviewer_izoN · 2024-03-12

**Summary Of Contributions:**

This paper studies the concept of ablation, which is an important topic for privacy and copyright protection. Specifically, this paper investigates image generation models. This paper reproduced results from existing work by Kumari et al., 2022. Additionally, the authors have introduced a new variant of concept ablation, the trademark ablation. It aims to remove well-known brands/trademarks from the generated images. Throughout several observational analyses, this paper demonstrated that trademark ablation can successfully remove the logo while not affecting other similarly generated images.

**Audience:**

Yes

**Claims And Evidence:**

Yes

**Requested Changes:**

- Typo of the single quotation marks. In both of the abstract and the Section 1.
- It would be great to use Section XXX rather than just XXX for the in-text references. E.g., in the paper outline. This could improve the readability.

**Strengths And Weaknesses:**

Strengths

- The concept of ablation is one of the tools that can be used for privacy and copyright protection. It is an important topic for ensuring the generated images do not incur copyright infringement. The concepts of ablation and trademark ablation studied have shown very promising results. It could have a large impact on the existing practices of releasing image-generation models to the public. The model owner can use the ablation for privacy and copyright protection.
- The experimental results are very promising, and observational analyses are very straightforward.

Weaknesses

- It might be better to include more numerical analysis. For example, a comparison between Eq (1), Eq (2), and Eq (4) for these observational analyses.
- Minor typos.

---

### Review · Reviewer_8vNy · 2024-03-26

**Summary Of Contributions:**

This paper extends the study of concept ablation within pre-trained diffusion models, which is done in Kumari et al. (2022). This paper first produces the claims made in the prior working then proposes a new variant of concept ablation called trade-mark ablation, which achieves better visual performance compared to instance ablation. The authors also showed the limitations of existing approaches against leakage-inducing prompts.

**Audience:**

Yes

**Broader Impact Concerns:**

The broader impact is addressed in the paper and is sufficient.

**Claims And Evidence:**

Yes

**Requested Changes:**

1. Add description of the notation in the paper. It is hard to parse the technical part without having a proper knowledge about the notations.
2. Record more data points in terms of different training steps so that we can have understandings about the trend with finder details.
3. Discuss about the possible reasons behind the limitations of existing approaches and ideally, can also propose some (tentative) mitigation solutions. I would suggest removing the reproducibility results to allow space for more interesting results.

**Strengths And Weaknesses:**

Strengths:
1. This paper showcases the limitations of existing concept ablation studies against leakage inducing prompts.

Weaknesses:
1. The experiments are only demonstrated on a few experimental data points and hence, as an empirical paper, the main messages may not be well supported with the limited empirical evidence.
2. There is a lack of understanding on how we may improve existing concept ablation approaches. It is also unclear what is the contribution of reproducing the claims made in prior work. I think more interesting results are ones that point out the limitations of previous works and then also proposing some tentative solutions.
3. There is a lack of understanding on how to qualitatively view the CLIP scores presented in the paper. Is below certain CLIP score threshold considered well-enough for practical applications?

---

### Review · Reviewer_YwiN · 2024-05-02

**Summary Of Contributions:**

The paper under submission is largely a work to reproduce previous results using various proposed ablation concepts.
The second contribution is to introduce a new ablation technique called trademark ablation. They also provide the limitations of the models and also study the model's behavior in response to ablation leakage-inducing prompts.

**Audience:**

Yes

**Broader Impact Concerns:**

There are none.

**Claims And Evidence:**

Yes

**Requested Changes:**

Nothing to add here.

**Strengths And Weaknesses:**

This is essentially a reproduction study, which I think is a good thing for science. They also introduce a new ablation concept. They consider three reproduction results: when the style of the artist is to be removed, when a specific instance or event has to be removed, and when memorization based image generation.
They have a clear evaluation plan and metric. They present their work as it is without trying to conflate their finding. I find this as a good welcome even pointing out the limitations of their method.

---

> ### Author Response · Authors · 2024-05-09
> **Reply to Reviewer YwiN**
>
> We are grateful for your insightful review and for recognizing the importance of our work in the context of reproduction studies. Thank you for acknowledging the significance of our findings and for highlighting the transparency in addressing the limitations of our method.

---

### Decision · Action_Editor_gCMY · 2024-06-07

**Recommendation:** Accept as is

**Comment:**

The paper is a valuable reproducibiltiy study with some additional insights and as such clearly meets the bar for TMLR.

**Audience:**

Ablation is a topic that has generated significant attention and so I don't see any issue with relevance for the TMLR audience.

**Claims And Evidence:**

The paper is primarily a reproducibility study and there is no reason to doubt any of the claims made in the paper.  One reviewer noted that the breadth of experiments is a bit limited, but there are no specific claims that appear to be lacking support.